# Telehealth Readiness of Healthcare Providers during COVID-19 Pandemic in Saudi Arabia

**DOI:** 10.3390/healthcare11060842

**Published:** 2023-03-13

**Authors:** Fatchima Laouali Moussa, Mahaman L. Moussa, Homood A. Alharbi, Tagwa Omer, Hussain Ahmad Sofiany, Yahia Ahmad Oqdi, Bandar Hammad Alblowi, Sally Hammad Alblowi

**Affiliations:** 1College of Nursing, Princess Nourah Bint Abdulrahman University, Riyadh 11564, Saudi Arabia; 2College of Nursing, King Saud University, Riyadh 11451, Saudi Arabia; 3College of Nursing, King Saud Bin Abdulaziz University for Health Sciences, Jeddah 21423, Saudi Arabia; 4Ministry of Health, Ohud Hospital, Medina 42394, Saudi Arabia; 5Ministry of Health, Medina 3843, Saudi Arabia; 6Ministry of Health, Dorrat Al Madina, Medina 42376, Saudi Arabia

**Keywords:** telehealth, readiness, COVID-19, healthcare providers, Saudi Arabia

## Abstract

Objective: This study aimed to assess and explore the telehealth readiness of healthcare providers in Saudi Arabia. Methods: This descriptive cross-sectional survey was conducted in a government healthcare facility in Saudi Arabia between August and October 2020. The Telehealth Readiness Assessment (TRA) tool was used. Results: A total of 372 healthcare providers participated in this study. Their mean age was 35.5 years (SD = 10.46). The majority of respondents were female (65.6%), nurses (68.0%), married (60.2%), and non-Saudi nationals (64.2%). The analysis shows that healthcare providers generally had moderate-to-high telehealth readiness. Of the five domains, financial contributions had the lowest rating among nurses and physicians, 63.4% and 66.1%, respectively. Gender (β = 7.64, *p* = 0.001), years of experience in the organization (β = 11.75, *p* = 0.001), and years of experience in the profession (β = 10.04, *p* = 0.023) predicted the telehealth readiness of healthcare providers. Conclusion: The telehealth readiness of healthcare providers in Saudi Arabia showed moderate to high levels. The COVID-19 pandemic poses a catastrophic threat to both patients and healthcare providers. Assessing telehealth readiness should include both patients and healthcare provider factors. A better understanding of the factors of organizational readiness, particularly healthcare providers, could help avoid costly implementation errors.

## 1. Introduction

The World Health Organization (WHO) declared the coronavirus disease 2019 (COVID-19) outbreak to be a pandemic in March 2020 [1]. COVID-19 has infected millions of people and caused thousands of fatalities in 215 countries, such as the United States, Spain, the United Kingdom, Italy, and Saudi Arabia [2]. According to mortality data, healthcare provider mortality had an increase of approximately 68.83% in 2020 [3]. The COVID-19 pandemic brought enormous pressure and changes to the general population, such as fear of becoming infected, uncertainty, confusion, and a sense of urgency [4]. In addition, the unprecedented situation caused by COVID-19 led to various psychosocial stressors and significant changes in daily routine in the public and in healthcare providers [5,6].

In response to the COVID-19 pandemic, different countries implemented several measures to suppress the rapid spread of the virus. These included quarantines and lockdown measures, suspension of flights, avoidance of social gatherings, and suspension of classes [7]. The COVID-19 outbreak and these preventive measures have had significant repercussions for the general population and healthcare providers. The COVID-19 pandemic has also shown how vulnerable the healthcare systems of different countries are. Moreover, the COVID-19 pandemic influenced hospital operations, such as routine patient care practices, which placed extraordinary demands on the healthcare system [8], for example, enhanced isolation practices, increased personal protective equipment (PPE) requirements, limitation and cancellation of elective procedures or surgeries, delay and disruption of treatment, and the modifying or prolonging of treatment intervals [8,9].

As a result of all the measures that have been implemented to prevent the spread of COVID-19, the healthcare sector has quickly adapted to meet the needs of patients. Telehealth is one of the adaptive actions to aid in patient access and care [10,11]. Telehealth is a type of healthcare service that uses information and communications technology to render long-distance clinical healthcare [11,12]. Telehealth involves all aspects of remote healthcare, such as clinical healthcare using telemedicine, patient and professional health-related education, public health, and health administration [13]. In the past two decades, telehealth provided care in difficult areas or specific populations, such as rural and prison populations [14]. The trend of using telehealth had been slowly but steadily increasing before the COVID-19 outbreak. Telehealth came into great focus to limit the spread of COVID-19.

Several studies examined the various benefits of using the technology of telehealth for different outcomes. For example, telehealth in hospitals improves outcomes in high-risk patients in rural areas [15]. Furthermore, telehealth reduces the risk of transmission of infectious agents, improves communication with providers, decreases travel time and waiting times, and improves medication adherence [15,16]. The results of a global survey to investigate the use of telemedicine before and after the COVID-19 pandemic revealed the top three barriers to implementing telemedicine, including patients’ lack of technological comprehension, patients’ lack of access to the required technology, and reimbursement concerns [17]. Another aspect that should be considered when using telehealth is the trust of people. The shift in information consumption to digital platforms, particularly social media sites, has accelerated the spread of health misinformation and contributed to mistrust in the healthcare system [18]. Telehealth in Saudi Arabia is a new concept established by the Ministry of Health as an e-Health strategy using telemedicine to improve the quality of care and services for patients in different rural areas [19]. Telehealth is increasingly used in every aspect of healthcare, and the success of telehealth implementation depends on the healthcare providers’ acceptance of and readiness for this new technology.

While the use of telehealth has been proven by several studies and the global experience of the COVID-19 outbreak, it has been indicated that timely and appropriate technology may play a considerable role in preventing the spread of COVID-19. Although it provides promising efforts and opportunities to reduce and protect patient health during the COVID-19 pandemic, there are still questions about whether healthcare providers are ready to adopt this new system or technology. Thus, this study aims to assess and explore the readiness of healthcare providers in Saudi Arabia concerning the availability of telehealth.

## 2. Methods

### 2.1. Design

This descriptive cross-sectional survey was designed to assess the telehealth readiness of healthcare providers during the COVID-19 pandemic in Saudi Arabia.

### 2.2. Sample and Setting

The current study used a convenience sampling strategy among healthcare providers working in public healthcare facilities in Madinah, Saudi Arabia. The inclusion criteria of the study included the following: participants had to be healthcare providers currently working in 10 public hospitals in Madinah, Saudi Arabia. The sample size was calculated using the G power program, software version 3.1.9.7. The sample size was set to a minimum of 293 healthcare providers, at 0.05 effect size, probability level of 0.05, and statistical power of 80% for seven predictors in multiple regression [20]. A total of 400 healthcare providers were randomly invited to participate in this study, and 372 complete questionnaires were received. The study was conducted between August and October 2020. Since the onset of the pandemic, the Ministry of Health has mandated strict precautionary measures and treatment guidelines for the effective management and control of COVID-19. The study was approved by the Ethics Committee of The Institutional Review Board General Directorate of Health Affairs in Madinah (IRB No. 567, H-03-M-084).

### 2.3. Procedure

The researchers used an online questionnaire compiled with Google Forms and distributed it to participants using email or WhatsApp with a link to the web page containing the informed consent form and a description of the study objectives. Participants were contacted using contact information and referrals from supervisors and department heads. All participants were reassured that their participation would not have affected their job status or work process. 

### 2.4. Instrument

The research questionnaire used in this study comprised two main parts. Part 1 comprised the demographic characteristics of the participants, such as age, gender, marital status, monthly income, years of experience in the hospital, and years of experience in their profession. The second part was the Telehealth Readiness Assessment (TRA) tool by Pollack et al. (2019), which aims to assess the readiness of healthcare providers for telehealth [21]. The tool focuses on the level of readiness in areas offering telehealth services, areas that need improvement, and prioritized improvement. The TRA tool has five key domains associated with the successful implementation of telehealth, such as core readiness, financial considerations, operations, staff engagement, and patient readiness. The instrument uses a 4-point Likert-type scale (0 = not applicable, 1 = no/unsure, 2 = somewhat, and 3 = definitely). The interpretation of the raw scores was carried out considering the instrument manual, where ≤50% means low-level readiness, between >50% and <75% means moderate readiness level, and >75% indicates high readiness level.

### 2.5. Statistical Analysis

Data analysis was performed using IBM SPSS Statistics V.23. Descriptive statistics were used for analyzing variables. Analysis of variance (ANOVA) and independent *t*-test were used to identify correlations between the healthcare provider characteristics and telehealth readiness. Multiple linear regression (enter method) was employed, after checking the multicollinearity and normality of the data, to identify which variables could explain the telehealth readiness of healthcare providers. The level of acceptable significance was set at *p* < 0.05.

## 3. Results

A total of 372 healthcare providers participated in this study. Their mean age was 35.5 years (SD = 10.46). The majority of respondents were female (65.6%), nurses (68.0%), married (60.2%), and non-Saudi nationals (64.2%). Fifty percent of the participants had a monthly income of SAR 10,000 or more. Nearly half of respondents had been in the current organization for 5–10 years, and more than half of participants (57.0%) had been in their profession for 10 years or more. The detailed characteristics of the healthcare providers are presented in Table 1.

Regarding the telehealth readiness of healthcare providers to care for COVID-19 patients, the analysis shows that healthcare providers had moderate-to-high-level telehealth readiness in general (Table 2). Of the five domains, financial contributions had the lowest rating among nurses and physicians, 63.4% and 66.1%, respectively. As presented in Table 2, the rates of scheduling and workflows, and education and awareness varied according to the participant characteristics. The rates of scheduling and workflow were 66.7% and 69.8% in nurses and physicians, respectively, while in the staff engagement domain, the rates of education and awareness were 64.1% in nurses and 71% in physicians. The other dimensions showed no significant differences (see Table 2 for details).

Based on multiple linear regression analyses, gender (β = 7.64, *p* = 0.001), years of experience in the organization (β = 11.75, *p* = 0.001), and years of experience in the profession (β = 10.04, *p* = 0.023) predicted the telehealth readiness of healthcare providers. In other words, the statistical model revealed that high readiness was associated with being male and having numerous years of experience in the current organization and profession (Table 3).

## 4. Discussion

The study presents that healthcare professionals had moderate-level telehealth readiness during the COVID-19 pandemic. This result is parallel to that of a survey conducted on dentists in Saudi Arabia, who had moderate-level readiness for teledentistry [22]. The average level of readiness could be due to no specific courses being provided on telehealth or telemedicine for healthcare providers. In addition, a possible explanation may be that the study period coincided with the heights of the pandemic. In comparison, our result is slightly higher than that of the telehealth readiness of clinical nurses in China [23]. This survey showed a significant difference between the telehealth readiness of physicians and nurses in the domains of financial contributions, scheduling and workflows, and education and awareness. Financial contributions represent a vital component of telehealth implementation [24,25]. Identifying the costs and benefits of telehealth before implementation is a critical part of sustainability [26]. Healthcare providers need to understand the identifying cost and benefits of telehealth in their practice so that they can make informed decisions [26]. Additional training may increase the awareness and readiness of healthcare providers, particularly the benefits of telehealth and clinical practice site, and the availability of technological guidelines. In this survey, scheduling and workflows, and education and awareness were assessed in participants, and the results showed moderate-to-high-level readiness. The results are similar to those of a study conducted in Austrian professionals that assessed the readiness to use telemedicine technologies for diabetic patient management and found that 58.2% of them had average readiness levels [27]. This can be explained by the well-organized infrastructure at the clinical practice site and the availability of guidelines that promote information and communication technology tools for patient care. Since physicians and nurses are directly involved in the care of COVID-19 patients and the delivery of healthcare services, it is essential to implement measures to increase their readiness for telehealth and the integration of telehealth in their patient care. 

This study highlights significant factors, such as gender, years of experience in the current organization, and years of experience in the profession, that influence telehealth readiness. In general, the implementation of telehealth is poorly advanced among practitioners [28,29,30,31]. This finding aligns with those of other studies that indicate a positive correlation between years of experience in the current organization and professionals’ readiness [23]. In addition, the study showed that the attitude of participants significantly affected telehealth readiness and telehealth implementation [28]. Participants who had a favorable attitude towards telehealth were 2.4 times more likely to have high readiness levels than participants with an unfavorable attitude towards telehealth. Healthcare providers might primarily consider telehealth as additional work rather than support. For example, a shift from physical to virtual doctor’s appointments could currently cause rapid and poor migration to virtual care, with both patients and healthcare providers lacking the necessary equipment to participate in virtual care. This is consistent with the findings of a qualitative research study conducted in the United States on residents’ readiness to use telehealth that identified three themes, including knowledge gap related to telehealth, infrastructure, and a desire for easier access to healthcare [32].

There are some limitations to the current study that need to be considered when interpreting the results. The study design limits causality from being inferred between independent and dependent variables. The study was conducted by only using a newly validated questionnaire, which may have limited participant responses and comparisons with other studies. Future research studies should consider adding a qualitative approach to strengthen the findings. Furthermore, the study was only conducted in one region, which may affect the generalizability of the findings to Saudi Arabia and other countries. Future works could be better with the incorporation of settings other than public hospitals.

In conclusion, the telehealth readiness of healthcare providers during the COVID-19 pandemic in Saudi Arabia showed moderate-to-high levels. The COVID-19 pandemic poses a catastrophic threat to both patients and healthcare providers, and telehealth provides easy and safe access to high-value or -quality care for patients. Telehealth readiness needs to be systematically assessed, which may help identify barriers and opportunities for long-term success. Assessing telehealth readiness should include both patient and healthcare provider factors. A better understanding of the factors of organizational readiness, particularly healthcare providers, could help avoid costly implementation errors. The information generated in this study could provide organizations with baseline knowledge to develop and implement appropriate and relevant telehealth training programs.

## Figures and Tables

**Table 1 healthcare-11-00842-t001:** Demographics of the healthcare providers.

Variable	Total N = 372
Age	Mean 35.5 SD 10.46
20–29	133 (35.8)
30–39	137 (36.8)
40 and above	102 (27.4)
Gender	
Male	128 (34.4)
Female	244 (65.6)
Profession	
Physician	119 (32.0)
Nurse	253 (68.0)
Marital Status	
Single	148 (39.8)
Married	224 (60.2)
Nationality	
Saudi	132 (35.5)
Non–Saudi	240 (64.5)
Monthly income	
Less than SAR 10,000	183 (49.2)
10,000 or more	189 (50.8)
Years of experience in organization	
Less than 5 years	95 (25.5)
5 to 10 years	159 (42.7)
10 years or more	118 (31.7)
Years of experience in practicing profession	
Less than 5 years	92 (24.7)
5 to 10 years	68 (18.3)
10 years or more	212 (57.0)

**Table 2 healthcare-11-00842-t002:** Differences in ratings of healthcare providers regarding telehealth readiness.

Domain	Concept	Nurses	Physicians	*p*-Value
Core readiness	Need for telehealth	66.7	70.2	0.462
	Organizational leadership Buy-in	67.5	69.1	0.125
Financial considerations		63.4	66.1	0.042
Operations	Telehealth roles	69.5	71.0	0.727
	Scheduling and workflows	66.7	69.8	0.001
	Operation requirements	70.2	72.3	0.415
	Assessment approach	74.3	76.1	0.526
	Technology	69.2	70.1	0.727
	Physical space	71.5	73.0	0.232
Staff engagement	Education and awareness	64.1	71.0	0.001
	Innovators/champions	69.0	72.4	0.560
Patient readiness	Patient engagement	76.1	79.2	0.181
	Health literacy	77.4	79.6	0.424
Overall				

Note: *p* < 0.05 was considered significant.

**Table 3 healthcare-11-00842-t003:** Multiple linear regression factors associated with the telehealth readiness of healthcare providers.

Variable	B (95% CI)	*p*-Value
Age	0.66 (−1.18–2.52)	0.063
Gender	7.64 (−2.04–14.43)	0.001
Marital status	−5.20 (−8.92–−18.56)	0.650
Profession	−2.67 (0.04–0.32)	0.111
Nationality	−4.93 (−10.42–10.54)	0.688
Monthly income	3.54 (−10.27–7.35)	0.828
Years of experience in the organization	11.75 (−10.83–17.34)	0.001
Years of experience in the profession	10.04 (−11.15–19.24)	0.040

Note: β, standardized regression coefficient; SE, standard error; CI, confidence interval; *p* < 0.05 was considered significant.

## Data Availability

The data presented in this study are available on request from the corresponding author. The data are not publicly available due to the nature of this research, participants of this study did not agree for their data to be shared publicly, so supporting data is not available.

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
