# Peer review of "Telehealth Readiness of Healthcare Providers during COVID-19 Pandemic in Saudi Arabia"

_healthcare, 2023, doi:10.3390/healthcare11060842_

Round 1
Reviewer 1 Report
The article investigate an interesting subject
- The aim of the study was to assess the readiness of healthcare providers in Saudi Arabia concerning the availability of tele- 78 health.
I suggest adding in discussion
- Comparison to other studies treating telehealth during pandemic years
- Comparison to other studies using the same questionnaire.
- Highlight the original contribution of the study
- I have asked the authors to mention their original contribution.
- I have no indications regarding the methodology.- The results must be compared with literature data.
- The references are appropriate
Author Response
Dear Reviewers,
Thank you very much for your valuable feedback. We have addressed all your comments as per below. We have also indicated the changes in our manuscript with yellow font color.
We are hoping for your positive feedback.
Reviewer 1
- The aim of the study was to assess the readiness of healthcare providers in Saudi Arabia concerning the availability of tele- 78 health.
I suggest adding in discussion
- Comparison to other studies treating telehealth during pandemic years. Comparison to other studies using the same questionnaire. Highlight the original contribution of the study.
- I have asked the authors to mention their original contribution.
- I have no indications regarding the methodology.
- The results must be compared with literature data.
- The references are appropriate
Response to reviewer 1: We have provided additional discussion to compare our results with other studies. We have also highlighted the original contribution of this study under the conclusion.
For more details please see the revised version manuscript.
Reviewer 2 Report
The manuscript entitled "Telehealth Readiness of Healthcare Providers During COVID19 Pandemic in Saudi Arabia" focused on the importance of telehealth in the control of Covid19 pandemic and the factors that could intervene in the efficiency of this type of service. These types of studies are essential in each society, which help governments to discover deficiencies and take the solutions into account before any crises. I believe that this type of study should be carried out on a larger scale to go to a reasonable conclusion.
Regarding this manuscript, an issue that could be addressed in the introduction is the trust of people. At the beginning of Covid 19 pandemic, people did not have that much trust in telehealth services. But little by little, as a result of positive feedback or advertisements, this service was culturalized among people.
To improve the manuscript, please find my comments below:
- The manuscript needs to be polished grammatically. There are some grammatical mistakes.
1. there was no line number??
2. Replace contain with suppress "In response to the COVID-19 pandemic, different countries implemented several measures to contain the rapid spread of the virus"
3. This sentence "The use of telehealth has been a slow but steady rise globally not until the COVID-19 outbreak hit" should be modified as follows: The trend of using telehealth has been increasing slowly but steadily before the COVID-19 outbreak.
4. Note: the tense of verbs should be harmonized in a sentence. Furthermore, telehealth reduces the risk of transmission of the infectious agent, improves communication with providers, decreases travel time and wait times, and improves medication adherence.
5. The tool focused on the level of readiness on in areas offering telehealth services, and areas that needed improvement or prioritized improvement by importance.
6. The TRA tool has had five key domains associated with a successful implementation of telehealth, such as core readiness
7. If you chose to use symbols < or >, Just use them in all occasions in a sentence: you can use ≤ 50% instead of (50% or lesser), etc.
Interpretation of the raw scores was carried out considering the instrument manual, where ≤ 50% means low-level readiness; > 50% and > 75% means moderate readiness level; and >75% indicates high readiness level.
8. Correct the sentence as follows: The Scheduling and Workflow were 66.7% and 69.8% among nurses and physicians respectively.
9. the table should be modified: try to draw three columns as follows:
Note: the following information could be mentioned inside the parenthesis in front of the table caption (Total N=372, Mean 35.5 SD 10.46)
Variables Quality Respondents (%)
Age 20-29 133 (35.8%)
30-39
≥ 40
Gender Male
Female
Profession physician
Nurse
and other rows as the same as the above sample...
10. Table 2: what is the unit of nurses and physicians? is it %?
11. The overall row in table 2 should be removed
12: remove the subtraction sign before < : Note; p-<0.05 considered significant.
13. Remove your in the following sentence: Based on multiple linear regression analyses, gender (β = 7.64, p = 0.001), years of experience in the organization (β = 11.75, p = 0.001), and years of experience in your profession (β = 10.04, p = 0.023) predicted telehealth readiness of healthcare providers.
14. Remove the subtraction sign before < : Note Note: β, Standardized Regression Coefficient; SE, Standard Error; CI, Confidence Interval; p-<0.05 considered significant
15. This highlights: "The highlights the significant factors such as gender, years of experience in the current organization, and years of experience in the profession that influence telehealth readiness"
16. Healthcare providers might primarily consider telehealth not as an support but rather as additional work rather than support.
17. Please rewrite this sentence: Thus, telehealth implementation should emphasize both the patient and healthcare providers factors in medical decision making and the financial aspect of a business model.
Author Response
Dear Reviewers,
Thank you very much for your valuable feedback. We have addressed all your comments as per below. We have also indicated the changes in our manuscript with yellow font color.
We are hoping for your positive feedback.
Reviewer 2
The manuscript entitled "Telehealth Readiness of Healthcare Providers During COVID19 Pandemic in Saudi Arabia" focused on the importance of telehealth in the control of Covid19 pandemic and the factors that could intervene in the efficiency of this type of service. These types of studies are essential in each society, which help governments to discover deficiencies and take the solutions into account before any crises. I believe that this type of study should be carried out on a larger scale to go to a reasonable conclusion. – We have indicated in our study limitation that the study was conducted in one region in Saudi Arabia.
Regarding this manuscript, an issue that could be addressed in the introduction is the trust of people. At the beginning of Covid 19 pandemic, people did not have that much trust in telehealth services. But little by little, as a result of positive feedback or advertisements, this service was culturalized among people. – added in introduction
To improve the manuscript, please find my comments below:
- The manuscript needs to be polished grammatically. There are some grammatical mistakes.
- there was no line number??
- Replace containwith suppress"In response to the COVID-19 pandemic, different countries implemented several measures to contain the rapid spread of the virus"- done
- This sentence "The use of telehealth has been a slow but steady rise globally not until the COVID-19 outbreak hit" should be modified as follows: The trend of using telehealth has been increasing slowly but steadily before the COVID-19 outbreak.– done
- Note: the tense of verbs should be harmonized in a sentence. Furthermore, telehealth reduces the risk of transmission of theinfectious agent, improves communication with providers, decreasestravel time and wait times, and improves medication adherence. – done
- The tool focusedon the level of readiness onin areas offering telehealth services, and areas that needed improvement or prioritized improvement by importance. – done
- The TRA tool hashadfive key domains associated with a successful implementation of telehealth, such as core readiness- done
- If you chose to use symbols < or >, Just use them in all occasions in a sentence: you can use ≤50%instead of (50% or lesser), etc. Interpretation of the raw scores was carried out considering the instrument manual, where ≤ 50% means low-level readiness; > 50% and > 75% means moderate readiness level; and >75% indicates high readiness level. – done
- Correct the sentence as follows: The Scheduling and Workflow were66.7% and 69.8% among nurses and physiciansrespectively.- done
- the table should be modified: try to draw three columns as follows:
Note: the following information could be mentioned inside the parenthesis in front of the table caption (Total N=372, Mean 35.5 SD 10.46)- done
Variables Quality Respondents (%)
Age 20-29 133 (35.8%)
30-39
≥ 40
Gender Male
Female
Profession physician
Nurse
and other rows as the same as the above sample...
- Table 2: what is the unit of nurses and physicians? is it %? – Yes
- The overall row in table 2 should be removed- done
12: remove the subtraction sign before < : Note; p-<0.05 considered significant.
- Remove yourin the following sentence: Based on multiple linear regression analyses, gender (β = 7.64, p = 0.001), years of experience in the organization (β = 11.75, p = 0.001),and years of experience in your profession (β = 10.04, p = 0.023) predicted telehealth readiness of healthcare providers. – done
- Remove the subtraction sign before < :NoteNote: β, Standardized Regression Coefficient; SE, Standard Error; CI, Confidence Interval; p-<0.05 considered significant- done
- Thishighlights: "The highlights the significant factors such as gender, years of experience in the current organization, and years of experience in the profession that influence telehealth readiness"- done
- Healthcare providers might primarilyconsidertelehealth not as an support but rather as additional work rather than support. – done
- Please rewrite this sentence: Thus, telehealth implementation should emphasize both the patient and healthcare providers factors in medical decision making and the financial aspect of a business model.- deleted
For more details please see the revised version manuscript.

Reviewer 3 Report
The manuscript titled “Telehealth readiness of healthcare providers during COVID-19 pandemic in Saudi Arabia" presents a descriptive cross‐sectional survey assessing the telehealth readiness of healthcare providers in Saudi Arabia.
The question is original and well defined; the results provide an advance in current knowledge; the results are interpreted appropriately, and they are significant; all conclusions are justified and supported by the results.
The article is written in an appropriate way; the data and analyses are presented appropriately.
The study is correctly designed and technically sound; the analyses are performed with the highest technical standards; the data are robust enough to draw the conclusions; the methods, tools, software, and reagents are described with sufficient details to allow another researcher to reproduce the results.
The conclusions are interesting for the readership of the Journal; the paper will attract a wide readership.
There is an overall benefit to publishing this work; the work provides an advance towards the current knowledge; the authors addressed an important long-standing question with smart experiments.
The English language is appropriate and understandable.
I would make two recommendations to the authors:
1. The SARS-Cov-2 epidemic on healthcare workers has led to an increase in mortality of up to +185% in some categories: the authors should insert these data (DOI 10.3390/healthcare10091684) in the introduction to contextualize the need for telehealth (lines 42 43)
2. In table 2, the last row ('overall') has no data. Correct this please
Therefore, my recommendation is to accept after minor revision.
Author Response
I would make two recommendations to the authors:
- The SARS-Cov-2 epidemic on healthcare workers has led to an increase in mortality of up to +185% in some categories: the authors should insert these data (DOI 10.3390/healthcare10091684) in the introduction to contextualize the need for telehealth (lines 42 43)
Added: From a mortality data, healthcare providers had an increase in mortality of approximately 68.83% in 2020 (Lupi et al., 2022).
- In table 2, the last row ('overall') has no data. Correct this please- deleted. done
Round 2
Reviewer 1 Report
The article can be published in the present form